# Leveraging Visual Tokens for Extended Text Contexts in Multi-Modal Learning

**Alex Jinpeng Wang**[1], **Linjie Li**[2], **Yiqi Lin**[1], **Min Li**[3],
**Lijuan Wang**[2], **and Mike Zheng Shou**[1✉]
[1]Show Lab, National University of Singapore    [2]Microsoft    [3]Central South University

## Abstract

Training models with longer in-context lengths is a significant challenge for multi-modal machine learning due to substantial GPU memory and computational costs. This exploratory study does not present state-of-the-art models; rather, it introduces an innovative method designed to increase in-context text length in multi-modality large language models (MLLMs) efficiently. We present Visualized In-Context Text Processing (VisInContext), which processes long in-context text using visual tokens. This technique significantly reduces GPU memory usage and floating point operations (FLOPs) for both training and inferenceing stage. For instance, our method expands the pre-training in-context text length from 256 to 2048 tokens with nearly same FLOPs for a 56 billion parameter MOE model. Experimental results demonstrate that model trained with VisInContext delivers superior performance on common downstream benchmarks for in-context few-shot evaluation. Additionally, VisInContext is complementary to existing methods for increasing in-context text length and enhances document understanding capabilities, showing great potential in document QA tasks and sequential document retrieval. The code is available at https://github.com/showlab/VisInContext.

## 1 Introduction

Large Language Models (LLMs), such as OPT, Mistral, and LLaMA-2 [4, 5, 6], have significantly advanced the field of Natural Language Processing (NLP). These advancements are partly due to the increased capability of LLMs to process long contexts, from 512 tokens [7] up to 16K tokens [6]. Building on these developments, recent multi-modal learning research [1, 8, 9, 10] has shifted focus from simple image-text pairs, like those in CC3M [11] and LAION-400M [12], to more complex and lengthy interleaved document datasets. Examples include web corpora like MMC4 [13] and the OBELICS [14] dataset, as well as PDF corpora like DocVQA [15].

However, training models on these complex datasets presents significant challenges due to the increased GPU memory and computational demands of extended contexts. For instance, while processing just 5M data items from MMC4 and 10M from the OBELICS dataset, OpenFlamingo-9B [9] resorted to sub-sampling text and processing only 256 tokens at a time, yet it still requires 32 80GB A100 GPUs for over three days. This highlights the need for more computation-efficient methods to handle long context lengths effectively.

In the domain of LLMs, two popular methods to extend context length are the use of memorizing banks [16] and novel self-attention mechanisms [17, 18]. These methods have inspired advancements in the multi-modality domain as well. For example, the Large World Model [19] introduces Ring Attention [18], and MA-LMM [20] employs memory banks to process long video understanding tasks. While these techniques have shown promise, our approach aims to increase in-context text length

---

✉: Corresponding Author.

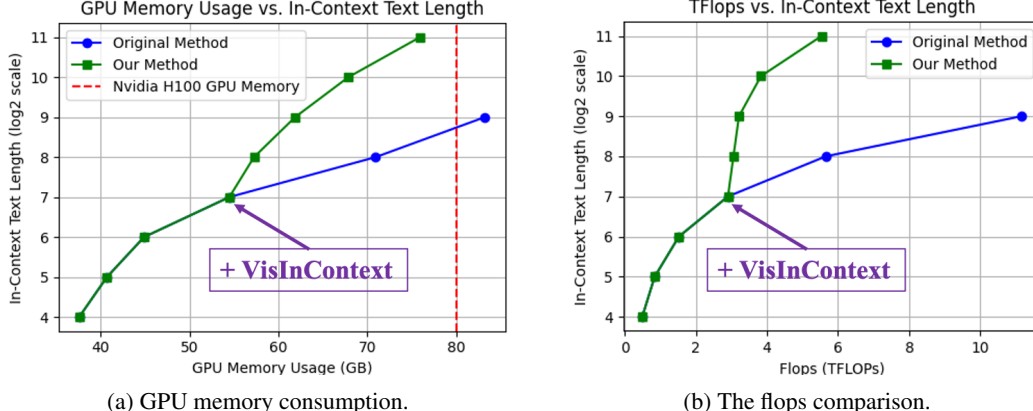

(a) GPU memory consumption.

(b) The flops comparison.

Figure 1: **VisInContext significantly increases the in-context text length from 256 to 2048 during pre-training on NVIDIA H100 GPU.** For our method, we incorporate VisInContext after 128 text tokens. We implement PyTorch Flamingo [1] models with different in-context length during pre-training. The language model is a 56B MOE [2] model loaded with 4-bit quantization and the batch size on each GPU is 32 with FP16. We train the model with DeepSpeed [3] Zero-2.

by leveraging the strengths of visual encoders in MLLMs. **We first observe that existing MLLMs usually exploit a much lighter visual encoders, compared to its text decoders**. For instance, Flamingo-9B consists of a 304.4M ViT-L/16 [21] as image encoder, and a 7.1B Chinchilla [1] model as the text decoder. Additionally, previous works [22, 23] have demonstrated that visual encoders trained on paired image-text data also **exhibit emergent OCR capabilities**.

Motivated by these observations, we propose Visualized In-Context Text Processing (VisInContext), a method that uses visual tokens to process extended textual contexts, which is complementary of existing methods in extending context length. Specifically, we **convert long textual content into images and use the visual encoders to extract textual representations**. In this way, we can efficiently and effectively enable models with much longer text contexts, as shown in Figure 1. With VisInContext, we show that the in-context text length can be increased by 7 times over the competing baseline. Additionally, we observe almost the same overall computation FLOPs even as in-context length extends significantly. Our extensive experiments will also show that VisInContext renders superior model performance on conventional in-context few-shot evaluations and document understanding, with much lower computational cost.

**Contributions.** In summary, our contributions are as follows: *i.* We introduce Visualized In-Context Text Processing (VisInContext), a novel method that increases in-context text length using visual tokens. VisInContext directly compresses text context at input-level, which is complementary to existing techniques with improved self-attention or memory banks. *ii.* We demonstrate that VisInContext is effective for both training and inference stage with much lower computational cost. *iii.* With extended text context brought by VisInContext, our model improves the average in-context few-shot performance from 55.8% to 57.8% over the competing baseline. *iv.* As a byproduct, our method also shows great potential in document understanding on popular document QA tasks and our newly proposed sequential document retrieval task.

## 2 Method

The goal of VisInContext is to process in-context text using visual tokens so that the model can handle long text context more efficiently. We primarily base our study on Flamingo-based architecture [1, 9, 14], as it has shown success in improving a model's ability to learn from long multimodal context that contains arbitrarily interleaved text and images.

### 2.1 Terminology

Before diving into model details, we define the following terms:

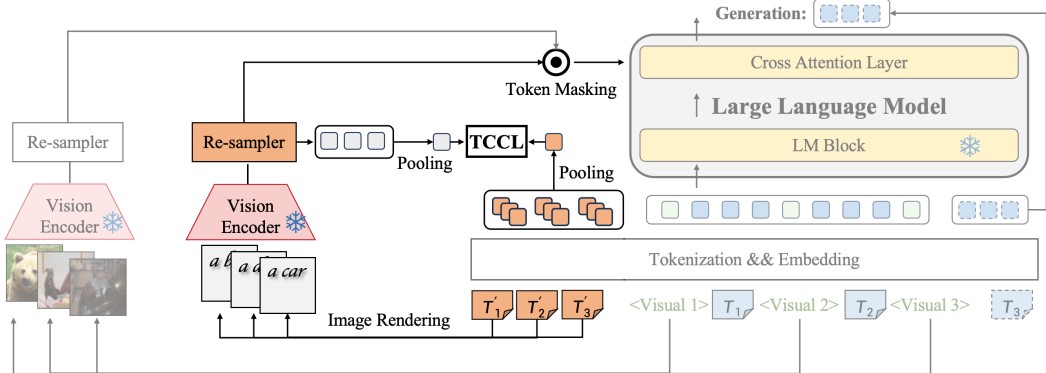

Figure 2: **VisInContext Pipeline**. The VisInContext pipeline builds upon the Flamingo model for in-context few-shot modeling (represented in gray). VisInContext processes interleaved image-text data by rendering portions of the in-context text into images. This approach maintains the *Text Token Length* of the model while allowing for a significantly extended *In-context Text Length*.

*In-context Text Length*: The **actual length of text tokens observed by the model within a document**.

*Text Token Length*: The length of the **text sequence input directly to the LLM**, corresponding to the token count of this sequence.

With VisInContext, the *In-context Text Length* is greater than the text token length, as part of the text is represented using visual tokens.

## 2.2 Overall Architecture

The implementation and architecture of VisInContext are shown in Figure 2. It is based on a dual-stream encoder model that integrates both visual and textual data. To effectively handle long interleaved data, we use a pre-sampling strategy as in Flamingo-style works [1, 9, 14]. We sample $m$ images $I_1, I_2, \ldots, I_m \in I$ with corresponding texts $T_1, T_2, \ldots, T_m \in T$. Tokens are concatenated in the form $< \text{visual}_1 >< \text{text}_1 > \ldots < \text{visual}_m >< \text{text}_m >$, where $< \text{visual} >$ is a single placeholder token. A random 256-token sequence is then sampled. However, since the overall length of a web document is generally much longer than 256 tokens (*In-context Text Length* $\geq$ *Text Token Length*), this sampling approach can lead to the omission of a lot of related text context.

To address this issue, we convert these omitted text context into visual signals by rendering them into images. We first concatenate all omitted text segments and divide them into $K$ parts to render text images, named $T_1^{'}, T_2^{'}, \ldots, T_m^{'} \in T'$. Both the original images and the text-rendered images are then processed through a shared frozen vision encoder. Then, we employ two learnable resamplers to extract a fixed number of tokens from both the raw and text-rendered image features, respectively. To facilitate the model to learn from rendered text images, we introduce two novel model designs, Token Masking mechanism and Text-Centric Contrastive Learning (TCCL). Token Masking allows the model to only read from text image tokens by masking the raw image tokens with masking ratio 1, which ensures that the model won't simply be ignoring the text images during training, hence can learn the association between the rendered text images $\{T_i^{'}\}$ and the text tokens $\{T_i\}$. TCCL aligns the visual text representation from the resampler with the embeddings extracted from text tokenizers in LLM, which reduces the gap between our visual text tokens and the text tokens the LLM is trained to perceive. With these designs, VisInContext not only reduces computational demands—as evidenced by a reduction in flops and inference time—but also improves the OCR ability, as we will show in our experiments.

## 2.3 Text Rendering

This module converts textual data into a visually rich RGB format, specifically rendering the text into an image size of $p_h \times np_w$, where $n$ is the number of patches. We employ the *HERSHEY* font at a size of 10px. On average, one 16x16 patch accommodates approximately 1.5 OPT text tokens. A 224x224 text image contains about 294 text tokens. Consequently, a visual encoder operating on this rendered text image requires only $1/3$ of tokens to encode an equivalent amount of text, compared

to the text tokenizer in language models. The vision encoder is quite lightweight ViT-L (340M) compared to language model MOE (56B), which makes **the processing of rendered text images significantly more efficient than directly inputting the text into a language model**.

## 2.4 Token Masking

In our initial experiments, we find that combining tokens from raw images and text images directly led to the network disregarding the text-image input. To address this issue, we introduce a Token Masking strategy to force the model to learn text semantics from visual inputs. During pretraining, the raw image and text image are first encoded into the same number of tokens after resampler, and then we mask the raw image tokens with a pre-defined probability. When masking out the raw image tokens, the model can focus on learning the association between rendered text images and the complementary text tokens. At inference time, we add the text-image tokens and image tokens together, to allow the model effectively leverage information from both sources.

## 2.5 Text-Centric Contrastive Loss (TCCL)

**Motivation.** Given that the vision encoder, typically a frozen Vision Transformer (ViT) [24], never observes rendered text images during pretraining, it may struggle to derive text semantics from pixels. To mitigate this issue, we introduce a new training objective, Text-Centric Contrastive Loss (TCCL). This objective aims to guide the resampler on rendered text images to interpret visual representations of text with a proficiency comparable to traditional text tokenizers, so that the textual semantics can be effective extracted from the rendered text images.

**Mechanism.** TCCL utilizes raw text token embeddings from the text tokenizer as soft supervision signals to supervise the resampler to learn text-centric representation. To reduce the global semantic gap between text image embeddings and text token embeddings, we first aggregate these embeddings with average pooling and then align them with TCCL. Intuitively, TCCL is designed to turn the joint of the vision encoder and resampler into a "visual" text tokenizer, as it promotes the text image embeddings to share a similar global semantic as the text token embeddings. The core of TCCL is formulated as a contrastive loss:

$$\mathcal{L}_{ij} = -\log \left( \frac{\exp(\text{sim}(f_{v_i}, f_{t_j})/\tau)}{\sum_{k=1}^{N} \exp(\text{sim}(f_{v_i}, f_{t_k})/\tau)} \right) \tag{1}$$

Where $\mathcal{L}_{ij}$ denotes the contrastive loss for comparing the $i^{th}$ text image against the $j^{th}$ text, $f_{v_i}$ and $f_{t_j}$ represent the feature embeddings of the $i^{th}$ text image and $j^{th}$ text, respectively. $\tau$ is a parameter that control the sharpness of the output distribution. Note that $f_{v_i}$ and $f_{t_i}$ are different features extracted from the same text, as the $i^{th}$ text image is a direct rendering of the $i^{th}$ text.

# 3 Experiment

## 3.1 Experimental Setup

**Pretraining.** We validate VisInContext with Open-Flamingo [9] and CosMo [25]. To enhance computational efficiency, all models utilize float16 precision. For the 56B MOE [2] model, we employ DeepSpeed's [3] Zero-2 stage with CPU offloading and further optimize the model by quantizing it to 4-bit precision [1]. We also use Flash Attention [17] to further improve memory efficiency. For all other experiments, we train the model using DeepSpeed Zero-2 without CPU off-loading. The Open-Flamingo 9B baseline is based on Mistral7B [5].

Our pretraining dataset includes a 180M subset of DataComp1B [26], MMC4 [13], the OBELICS [14] dataset, and OCR Rendered Text [27]. (More details are provided in the Appendix B.1) For each input document or image-text pair, we render a text sequence into an image with a fixed size of 16x8192 (512 patches) by default, with $p_h = p_w = 16$.

---

[1]The implementation is from https://github.com/TimDettmers/bitsandbytes.

| Method | Text | ICL Tokens↑ | Shots | VQA | | | | Caption | | Classi. | Mean |
|---|---|---|---|---|---|---|---|---|---|---|---|
| | | | | okvqa | textvqa | vizwiz | vqav2 | coco | flickr | HM | |
| **Open-Flamingo MOE [9]†** | Raw Text | 256 | 0 | 40.2 | 21.3 | 23.3 | 47.8 | 82.3 | 59.4 | 60.4 | 47.8 |
| | | | 4 | 42.5 | 22.2 | 32.2 | 49.8 | 90.5 | 63.5 | 63.8 | 52.1 |
| | | | 32 | **46.8** | 23.2 | 40.5 | 49.9 | **98.2** | 66.2 | **66.0** | 55.8 |
| **+VisInContext** | +Rendered Image | 2048 | 0 | 39.5 | 26.4 | 26.3 | 48.5 | 84.4 | 60.5 | 62.2 | 49.7 |
| | | | 4 | 44.3 | 28.9 | 32.0 | 50.3 | 94.2 | 65.3 | 65.5 | 54.4 |
| | | | 32 | 46.3 | **31.2** | **41.2** | **51.0** | **101.3** | **68.4** | 65.2 | **57.8** |

Table 1: **Increasing in-context text length with VisInContext significantly improves performance on multi-modality downstream tasks.** The model is pre-trained with a 56B MOE model. ICL stands for in-context text length. HM is short for hatefulmemes. With VisInContext, we increase the ICL from 256 to 2048, leading to clear improvements over the baseline. † indicates our implementation.

| Method | Text Source | Text Tokens | T-Shots | VQA | | | | Caption | | Mean |
|---|---|---|---|---|---|---|---|---|---|---|
| | | | | okvqa | textvqa | vizwiz | vqav2 | coco | flickr | |
| **Open-Flamingo9B Baseline [9]** †| Raw Text | 10 | 0 | 18.1 | 14.8 | 21.5 | 26.5 | 40.1 | 32.1 | 25.5 |
| | | 62 | 4 | 23.8 | 18.1 | 23.7 | **40.5** | 57.5 | 35.3 | 33.2(7.7↑) |
| | | 426 | 32 | **25.2** | 16.4 | **25.5** | 34.6 | 66.1 | 38.5 | **34.4(8.9↑)** |
| **+VisInContext** | Rendered Image | 10 | 0 | 16.2 | 16.8 | 15.4 | 30.6 | 42.3 | 33.5 | 25.8 |
| | | 10 | 4 | 17.2 | 21.8 | 19.7 | 35.2 | 52.4 | 35.2 | 30.3(4.5↑) |
| | | 10 | 32 | 21.3 | **22.6** | 21.5 | 38.8 | 60.3 | 37.0 | **33.6(7.8↑)** |

Table 2: **VisInContext effectively incorporates in-context text with visual tokens, demonstrating significant performance improvements with consistent token usage**. Here, T-shots refer to text-only in-context examples. Tokens indicate the length of the input to the LLM. Text source describes the preprocessing method for in-context examples. † denotes our implementation on 180M pretraining data.

**Downstream Evaluation.** Our objective is to demonstrate that in-context length can be extended using visual tokens, thereby enhancing the understanding of complex multimodal documents. Consequently, we focus primarily on tasks related to long-context understanding.

To evaluate the long-context understanding ability, we adopt the few-shot evaluation setting in Flamingo [1]. We report answer accuracy on the OK-VQA [28], TextVQA [29], VizWiz [30], and VQAV2 [31]. Additionally, we assess performance on captioning tasks using COCO [32] and Flickr30K [33]. Moreover, we also propose a setting named **text-only in context few-shots** to explore text-only in-context evaluation. For this setting, we use in-context sampling without visual input to generate long-context inputs and the visual input is not observed by the model.

In order to illustrate the impact of having long in-context text, we evaluate the model for document understanding on DocVQA [15] and OCR VQA [34]. Lastly, we introduce a new task, sequential multimodal document retrieval. This dataset is based on the existing interleaved OBELICS [14] dataset. Further details are provided in the Sec. D of the appendix.

### 3.2 In-context Few-shot Evaluation

**Impact of Extended In-Context Text Length.** Interleaved document datasets typically contain long texts. For instance, the OBELICS [14] dataset has an average token length of 815 tokens per document. Due to GPU memory constraints, Flamingo-like models [14, 9] only sub-sample 256 tokens during pretraining, which leads to a significant loss of context information. We compare the baseline model pre-trained with 256 tokens, against our method with an increasing *In-context Text Length* to 2048 tokens. Table 1 shows a clear advantage of VisInContext. For example, on TextVQA, accuracy improves from 23.2% to 31.2% with 32-shot. Similarly, the average model performance across all datasets show an increase from 55.8% to 57.8%. These findings demonstrate that **VisInContext effectively increases the *In-context Text Length* to improve multi-modality understanding**.

**Few-shot Evaluation with Text-only In-context Examples.** As downstream tasks often differ in format from pretraining data, several works [1, 9, 14] have tested the few-shot abilities of models

| Method | Text Source | DocVQA | | OCR VQA |
|---|---|---|---|---|
| | | val | test | |
| **Open-Flamingo-9B Baseline [9]** | Raw Text | 45.3 | 48.2 | 51.5 |
| **+VisInContext** | Rendered Image | **48.5(3.2↑)** | **52.2(4.0↑)** | **58.4(6.9↑)** |

Table 3: **VisInContext clearly boosting the baseline on document understanding tasks.**

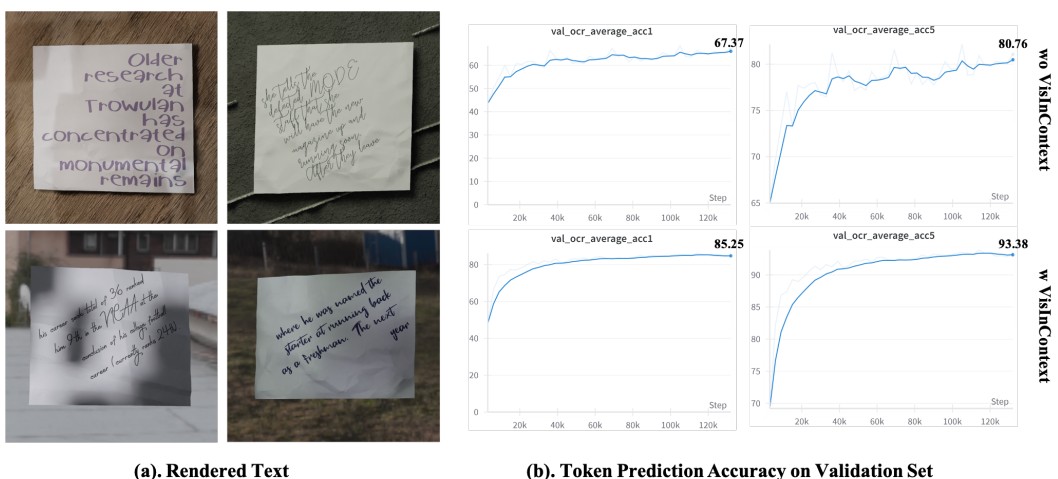

**(a). Rendered Text**  **(b). Token Prediction Accuracy on Validation Set**

Figure 3: **VisInContext significantly improves the OCR ability of LLM**. We present the Rendered Text [27] images and the corresponding next-word prediction accuracy on the validation set. Using the same pre-training steps, VisInContext achieves significantly better results in predicting words in visual images, even when the fonts are difficult to recognize.

using in-context examples. For instance, in the VQA dataset, a few question-and-answer pairs are provided as in-context examples with visual signals. However, for zero-shot evaluation, two question-and-answer pairs are added as in-context examples without visual signals in [1, 9, 14]. Follow the zero-shot setting, we examine the effect of having text-only in-context examples and extend it to multi-shot setting, by leaving out the corresponding images (See Appendix .E for more details). We compare model performance of the baseline Open-Flamingo 9B and our method under the same setting, where the differences lie in how these text-only in-context examples are processed. Specifically, Open-Flamingo directly takes in them as text tokens, while VisInContext takes in the corresponding rendered text images.

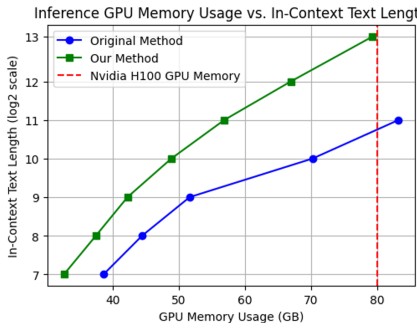

Figure 4: VisInContext extends the in-context text length of MOE based MLLM from 1k to 9k at inference stage.

Table 2 summarizes the results across four VQA benchmarks and two captioning benchmarks. Notably, compared to the text-only 0-shot setting, our VisInContext with 32-shot significantly is improved on all VQA and captioning benchmarks considered. Though the 32-shot performance of VisInContext is slightly lower than the competing baseline, we cut down the input tokens to the LLM from 426 to only 10 *Text Token Length*, which lead to significant reduction in the inference cost. These outcomes highlight two key points: *i*. **VisInContext can effectively understand text rendered in images**. *ii*. Text rendered as images can be comparably effective as raw text, when used as text-only in-context examples.

**Comparison on Inference Cost.** We then analyze the inference cost of VisInContext and compare to the baseline. Both models are based on a 56B MOE LLM with a batch size of one to explore the maximum manageable *In-context Text Length*. The results, shown in Figure 4, demonstrate that the *In-context Text Length* can be extended up to 9192 tokens for the 56B MOE model on 80GB H100 GPUs with our method at inference stage. This result

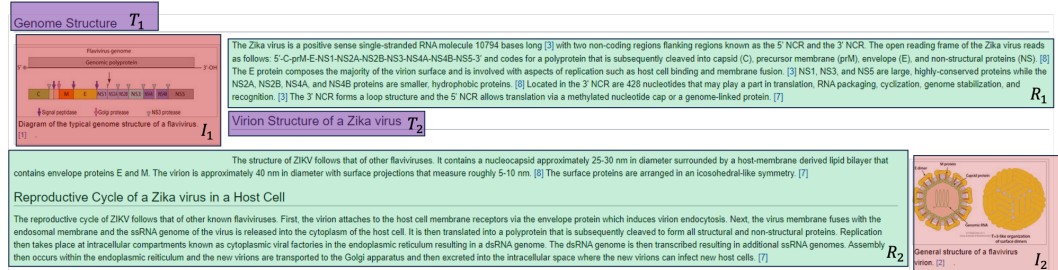

Figure 5: **Sequential multi-modal retrieval example.** The input sequence is $I_1, T_1, R_1, I_2, T_2, R_2$ that from interleaved document in OBELICS [14] dataset.

| Visual Input | Text Input | Surrounding Text Input | Seq-I | Seq-T |
|---|---|---|---|---|
| Raw Image | Raw Text | - | 16.3 | 64.8 |
| Raw Image | Raw Text | Raw Text | 18.9 | **67.5** |
| Raw Image | Raw Text | Rendered Text Image | **22.7** | 66.5 |

Table 4: **The model pretrain with VisInContext significantly improves sequence understanding ability.** We report the sequence retrieval result on OBELICS-Hybrid6.

highlights the efficiency and advantages of VisInContext, also show its potential in understanding very long document.

### 3.3 Document understanding

In this section, we evaluate the model on document understanding tasks. Unlike common vision-language tasks that usually short-form pairs, this task requires comprehension of long and complex document data. We evaluate our model on DocVQA and OCRVQA. All document images are of size $384 \times 384$. Following Pix2Struct [35], we finetune the model on DocVQA train data and report performance on the average normalized Levenshtein similarity (ANLS) metric.

Results in Table 3 show that our method significantly outperforms the baseline. For instance, we achieve a 6.9% improvement on OCRVQA. To further analyze why our method enhances document understanding, we present the validation accuracy of the LLM on the Rendered Text [27] dataset during pretraining in Figure 3. We observe a substantial improvement in next word prediction accuracy, with top-1 accuracy increasing from 67.37% to 85.25% (a 16% improvement) and top-5 accuracy rising from 80.76% to 93.38%. These findings indicate that the **LLM can effectively understand text embedded in visual signals with VisInContext**.

### 3.4 Sequential Multi-modal Retrieval

In order to further analyze the benefit of having long text context in multimodal modeling, we propose a new task – Sequential Multimodal Retrieval (SMR), based on document data from interleaved OBELICS [14] dataset. The document is composed of interleaved data, consisting of images and texts arranged in a meaningful sequence.

We show one sample in Figure 5 and define the input and output of this task as below: **Input:** Given a pair of content items, an image and a corresponding text $(I_1, T_1, R_1, I_2, T_2, R_2)$, from a document $D$. $I$ is Image, $T$ is the matched text and $R$ is the surrounding text. **Output:** The task is to retrieve the next image $I_2$ and the next text $T_2$ in the sequence. Named as Seq-I and Seq-T, correspondingly.

We sample the first 1K documents that contain data like $I_1, T_1, R_1, I_2, T_2, R_2$ from OBELICS [14] and named it as OBELICS-Hybrid6, which have at least three frames and three texts. (See Sec. E in appendix for more details.) This task encourages the model to leverage the contextual and semantic relationship in interleaved sequences to effectively predict and retrieve the subsequent pair.

To enable our model with retrieval, we follow CosMo [25] to add a simple contrastive head between visual embedding and language embedding from the middle layers. Recall that visual embeddings are either from raw images or rendered images or the addition of the two in our method. Table 4

| Method | Pretrain Text Source | Task DocVQA-val |
|---|---|---|
| **FuYu9B [8]**† | Raw-Text | 42.3 |
| **+ VisInContext** | +Rendered Image | **44.5 (2.2↑)** |

Table 5: **Pretraining with VisInContext helps on long-context understanding task for FuYu model**. † means our implementation on 180M data.

reports the results from our model with several input variants. We observe taking surrounding text input as rendered text image performs much better on the Sequence to image retrieval, while on par on Sequence to text retrieval, when compared with taking surrounding text input as raw text. These results further support the designs of VisInContext in the context of document understanding.

## 3.5 Extension to MLLM with Linear Embedding

Beyond utilizing the visual encoder, some works [36, 8] also employ linear embedding to extract visual features directly from raw images. To show the generality of our method, we also explore FuYu [8] model as a baseline and integrate VisInContext into the model. (See Sec. A in the appendix for more details.) As indicated in Table 5, our method is successful in improving the performances on DocVQA dataset that require long-context understanding.

| Text Image | Token Masking | TCCL | Ok-VQA | TextVqa | VizWiz | VqaV2 |
|---|---|---|---|---|---|---|
| | | | 11.5 | 15.3 | 8.7 | 24.2 |
| ✓ | | | 11.3 | 15.0 | 9.4 | 30.1 |
| ✓ | ✓ | | **17.8** | 18.3 | 15.3 | 33.5 |
| ✓ | | ✓ | 13.5 | 15.3 | 10.3 | 30.9 |
| ✓ | ✓ | ✓ | 17.2 | **21.8** | **19.7** | **35.2** |

Table 6: Ablation study of the component in our pipeline for text-only 4-shot example.

| Font Size | 4 | 6 | 8 | 10 | 12 |
|---|---|---|---|---|---|
| TextVQA | 15.4 | 17.2 | 18.5 | **21.8** | 20.3 |
| DocVQA | 39.8 | 42.5 | **45.6** | 44.3 | 36.2 |

Table 7: **Font size ablation.** We report the result on DocVQA val dataset.

| Dataset | 2 | 4 | 8 | 16 | 32 |
|---|---|---|---|---|---|
| TextVQA | **21.8** | 20.5 | 21.3 | 18.5 | 15.3 |
| DocVQA | **44.3** | 43.2 | 39.4 | 40.5 | 36.6 |

Table 8: **Font interval thresh ablation.** Larger thresh leads to few texts in general.

## 3.6 Ablation Study

**Ablations on Model Design.** We conduct ablation studies on the following modeling components to demonstrate their effectiveness: Text Image, TCCL, and Token Masking. Results are detailed in Table 6, which reveal two findings: 1. Token Masking is crucial for the model to learn from rendered text images. Without Token Masking, the model can only perform comparably to the baseline. Forcing the model to learn text semantics from rendered text images via token masking significantly improves model performance. 2. Utilizing TCCL with Token Masking yields better performance than using Token Masking alone.

**Ablations on Font Size and Interval Threshold.** As shown in Table 7, optimal performance varies with changes in font size. We found that adjusting the font size impacts performance similarly to altering the patch size—both methods effectively increase the contextual information within each patch. We prefer modifying the font size over the patch size because it allows for more intuitive adjustments. Our findings indicate that the model does not need a highly detailed understanding of each word to perform effectively.

Another important factor is the font interval threshold. As shown in Table 8, we observed that a too-large interval leads to inferior results. This is intuitive because a larger threshold results in fewer texts in the rendered text image.

# 4 Related Work

**Multimodal Language Models.** Current mainstream Multimodal Large Language Models (MLLMs) [37, 38, 22, 39, 40, 41] leverage the capabilities of Large Language Models (LLMs) [42, 6] due to their strong reasoning abilities, as demonstrated by recent advancements. These models typically adopt one of two primary designs for integrating visual information. The first approach involves the effective adaptation of visual representations, which are acquired via a separate visual encoder, into the text-based LLM framework like CLIP, GIT, and BLIP2 [22, 43, 37]. The representative method in this category incorporates visual representations into the language model using cross-attention, as seen in the Flamingo series models [1, 9, 14]. Along this line, recently some works like LLaVA [40], EMU2 [44], InternVL [45], DeepSeeker [10], and QWen [41] lead to superior results on multi-modality tasks with supervised finetuning on high-quality data. The second approach uses visual embeddings directly as input "tokens" for the LLMs, bypassing the traditional use of a separate visual encoder. This method processes visual patches with a linear layer and uses the resulting embeddings as direct inputs to the LLM, as implemented in models like ViLT [36] and FuYu [8]. This strategy omits the need for an additional visual encoder and simplifies the architecture.

In this work, we adopt the Flamingo [1] architecture as our main baseline for the following reasons: First, the Flamingo model emphasizes in-context few-shot learning ability and designs comprehensive few-shot evaluation strategies. Second, our focus is on extending the in-context text length during pre-training rather than on supervised fine-tuning.

**Enhancing Text Understanding through Visual Inputs.** Traditional text tokenization processes raw text efficiently, but it faces challenges such as vulnerability to spelling errors and limited cross-lingual transferability [46, 47]. These issues have prompted the exploration of tokenizer-free models, which aim to improve robustness and facilitate better cross-language applicability. For instance, a single spelling error can lead to entirely different tokens using traditional tokenization methods, impacting model performance.

Recent developments have seen innovative approaches like the Pixel model [46], which proposes processing text as an image using both an image encoder and an image decoder. This approach has sparked a series of studies that process not only textual data but also images, charts, and tables through a unified visual input system [35, 46, 48, 47]. These models are trained on a diverse array of visual data, such as webpage screenshots and user interface images, sourced extensively from the internet. They are specifically designed to handle visually-situated text in an end-to-end manner, offering the potential to support a wide range of applications.

**Long Context Modeling.** The challenge of incorporating more tokens into LLMs is an active area of research [49, 50]. Common approaches involve novel self-attention mechanisms [18, 51, 52] , compressed token [53, 54, 55] or memory banks [16]. Some works [56] exploit tensor parallelism or sequence parallelism to reduce memory costs. There also have some works focus on position embedding [57, 58]. In multi-modality research, closed-source models like Gemini [59] and GPT-4V [60] support long context inference up to millions of tokens. Open-source models such as MA-LMM for Long-Term Video Understanding [20] can process up to one hour of video using a long memory bank. The most relevant work Large World Model [19] extends token length using Ring Attention.

In contrast to these methods, our method utilizes off-the-shelf LLMs and compresses text tokens into visual tokens for efficient processing. Our method is complementary to these existing techniques and can be integrated with them to achieve lower computational cost and longer context length.

# 5 Conclusion and Limitations

This paper centers on multi-modality learning and addresses the in-context length limitations presented by heavy computational cost of LLMs in MLLMs. Our contribution is a novel and efficient method named VisInContext, which enables the model to perceive long text context as rendered text images. Comprehensive experiments show that VisInContext is effective on conventional in-context few-shot evaluations and document understanding, while being much more efficient.

One limitation of our method is, currently our method requires processing a fixed size image even for brief texts. In future work, we plan to dynamically reduce token counts with variable image sizes by retaining only non-empty tokens during pre-training. We aim to expand this method to additional tasks and encourage the community to further explore this direction.

## Acknowledgement

This research is supported by the National Research Foundation, Singapore under its AI Singapore Programme (AISG Award No: AISG3-RP-2022-030).

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

# A    Extending VisInContext to MLLM Using Only Linear Embedding

In Multimodal Large Language Model (MLLM), besides using visual encoders, there are several works [36, 8] that utilize only linear embeddings to encode visual information. To demonstrate the generality of our method, we first extend it to the FuYu [8] model in this section. We evaluated our methodology using the FuYu [8] architecture, a prominent model that leverages visual information through simple linear embeddings. This approach utilizes a large language model framework without the need for a visual encoder, instead employing linear embedding to process visual information.

## A.1    Methodology

Our system architecture, based on FuYu [8], incorporates a single decoder where both visual and textual inputs are converted into token embeddings and processed by the same Transformer structure. Inputs are divided into two segments: the first is a sequence of image patches forming a screenshot and image patches from text image, and the second a sequence of textual tokens that contextualize the screenshot. The configuration of this uni-modal approach is depicted in Figure 6.

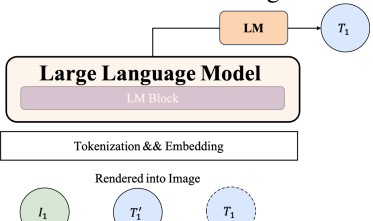

Figure 6: The main pipeline is based on Fuyu [8]. What's different is we introduce an additional text image. During pre-training, the rendered text image and original image is also alternatively. The DCSE is preserved. We show one image-text pair here for simplicity.

**Input Format.** The model inputs are formatted such that a text sequence of $m$ tokens is divided into screenshot segment ($m_s$) tokens and text segment ($m_t$) tokens, each comprising 256 tokens. The screenshot dimensions are defined as $p_h \times p_w$ pixels. Special tokens are integrated to guide the model's understanding of segment differentiation.

The rendering strategies are consistent with those employed in our visual encoder-based methods.

**Architecture.** The architecture follows the FuYu model [8], a widely used framework among visual encoder-free multi-modal language models (LMs), utilizing a 9 billion parameter model. Image patches are transformed into embeddings through a linear projection, while textual inputs utilize corresponding word embeddings. These embeddings are subsequently processed together in the Transformer blocks. To keep contrastive loss, we compute the average token metric after 3rd layer.

## A.2    Evaluation Settings

The primary objective of this experiment is to assess whether autoregressive screenshot language models (LMs) can accurately interpret text within screenshots using only linear embedding for the rendered text image. In this setup, the screenshot LM processes 256 text tokens derived from the screenshot context along with an additional 25 text tokens.

## A.3    Training Settings

Our training setup follows the Flamingo model, using image-text data sourced from DataComp [26]. For interleaved data, since FuYu does not support interleaved input, we sample one frame and a fixed length of text during pretraining. The model is trained with DeepSpeed Zero 2 optimization [3] and uses fp16 data type. We initialize the Language Model from Persimmon-8B weight.

# B    Pretraining & Downstream Task Evaluation details

## B.1    Pretraining Data Details

The associated data statistics for pretraining, presented comprehensively in Table 9, mainly include Datacomp [26] subset, MMC4 [13], Obelics [14] and Rendered Text [27].

| Data Type | Dataset | Sample |
|---|---|---|
| Image-Text | DataComp1B [26] | 108M |
| Interleaved Image-Text | MMC4 [13] | 30M |
| | Obelics [14] | 30M |
| Synthetic Data | Rendered Text [27] | 12M |
| Total | - | 180M |

Table 9: **Statistics of the Pre-training Dataset: Subsets are random sampling.**

The comparison between our method and Open-flamingo baseline, utilizing equivalent-scale pre-training data, consistently demonstrates the superior performance of our approach across diverse tasks.

| | | VisInContext 3B | VisInContext 9B | VisInContext 57B |
|---|---|---|---|---|
| **Model** | Language Model Backbone | OPT-IML-1.8B [61] | Mistral-7B [5] | MOE 56B [2] |
| | Vision Model Backbone | openai/clip -vit-large -patch14 | openai/clip -vit-large -patch14 | laion/CLIP-ViT -H-14-laion2B -s32B-b79K |
| | Cross-Layer Interval | 2 | 4 | 4 |
| **Training** | Text Sequence Length | 128 | 128 | 128 |
| | ICL Text Length | 2048 | 2048 | 2048 |
| | Effective Batch Size | 3072 | 1536 | 768 |
| | Max Training Steps | 200K | 200K | 500K |
| | Weight Decay | 0.1 | 0.1 | 0.1 |
| | Optimizer | adamw(0.9, 0.999) | adamw(0.9, 0.999) | adamw(0.9, 0.999) |
| | Gradient Clipping | 1.0 | 1.0 | 1.0 |
| **Learning Rate** | Initial Max | 5e-5 | 3e-5 | 3e-5 |
| | Decay Schedule | Cosine | Cosine | Cosine |
| | Linear warmup Steps | 5000 | 5000 | 5000 |

Table 10: **The hyperparameters used in pre-training for three distinct VisInContext variations**. The learning rate and batch size is smaller for sine the GPU memory limitation is 32GB.

## B.2  Hyperparameter Configuration

In this subsection, we outline the essential training details required for reproducibility. Our experiments included three different model sizes, with larger models requiring smaller batch sizes due to GPU memory limitations. We employed DeepSpeed ZeRO-2 optimization with fp16 precision and adjusted gradient accumulation steps to match the data type count. Comprehensive results are presented in Table 10.

The text input sequence length is 128 as default. Since we adopt VisInContext to increase the in-context length, the ICL length can be increased to 2048, around 15 text images. The $\tau$ for TCCL is 0.07.

## B.3  Parameter Details

The Flamingo [1] baseline include Resampler, Language Model, Visual Encoder, Cross-attention. Both the Language Model and Visual Encoder are frozen during pretraining. We mainly train the Gated Cross Attention layer and Resampler layer.

| Model | Language | Vision | Gated Cross Attention | Resampler |
|---|---|---|---|---|
| Flamingo-9B | 7.1B | 428M | 0.8B | 194M |
| Flamingo-9B Baseline † | 7B | 307M | 0.35B | 194M |
| MOE Baseline† | 56B | 307M | 0.5B | 194M |

Table 11: Parameter counts for each component in MLLM. † means our implementation.

| Method | Text Source | Parameter | Text Tokens↓ | T-Shots | VQA | | | | Mean |
|---|---|---|---|---|---|---|---|---|---|
| | | | | | ok-vqa | textvqa | vizwiz | vqav2 | |
| **OPT-1.3B [4]** | Rendered Image | 1.3B | 10 | 0 | 11.2 | 15.8 | 5.4 | 33.6 | 16.5 |
| | | | 10 | 4 | 17.2 | 21.8 | 7.8 | 33.2 | **20.0 (3.5↑)** |
| | | | 10 | 32 | 21.3 | 22.6 | 11.5 | 35.8 | **22.8(6.3↑)** |
| **MPT** | Rendered Image | 7B | 10 | 0 | 28.5 | 23.2 | 24.4 | 37.7 | 28.5 |
| | | | 10 | 4 | 30.1 | 23.2 | 28.4 | 40.3 | **30.5(2.0↑)** |
| | | | 10 | 32 | 32.5 | 25.4 | 30.3 | 41.8 | **32.5(4.0↑)** |

Table 12: **VisInContext performs well over different Language Models**.

## B.4 Extension to Other Language Models

Our research extends beyond the Mistral model, incorporating other language models such as OPT and MoE. The comparative results are summarized in Table 12. We noted marked improvements in performance across all models with the use of more in-context examples. This indicates that VisInContext's effectiveness is not highly dependent on the specific language model used, **showcasing the broad applicability and robustness of our methodology**.

## C Document Understanding Example

In this experiment, we present examples to demonstrate our method. As shown in Figure 7, we provide samples from the validation sets of DocumentVQA [15] and ChartQA [62].

Using VisInContext, we observe that the method answers questions more accurately, even when the font is unclear. For instance, consider the first pdf image have low resolution.

## D Sequential Multi-modal Retrieval Details

**Data Collection** We retrieve 1,000 samples from the OBELICS dataset, each sample consisting of six segments in the fixed order: $I_1, T_1, T_2, I_2, T_3, T_4$. Each image has one matching text segment and additional surrounding text. We use relative positioning to indicate which text is matched and which is surrounding.

**Retrieval Details** To perform the retrieval task, we incorporate contrastive loss during pretraining, following the approach of CosMo [25]. We add a contrastive head for the uni-modality text and vision embeddings. Using the mean of the text and image features as a query, we retrieve the next image or text segment. This task tests the model's ability to handle long in-context text.

We compute the dot product similarity and rank the scores to determine the final result. When processing surrounding text as "Raw Text," we concatenate $T_1$ and the surrounding text $R_1$ directly to the LLM to obtain the text embedding. We then use the mean of this text embedding and the image embedding of $I_1$ to retrieve the next image or text.

For our method, we render the surrounding text $R_1$ into an image and use the sum of two resampler outputs as the image embedding. We compute the mean of this image embedding and the $T_1$ embedding and use this mean vector to retrieve the next image or text segment.

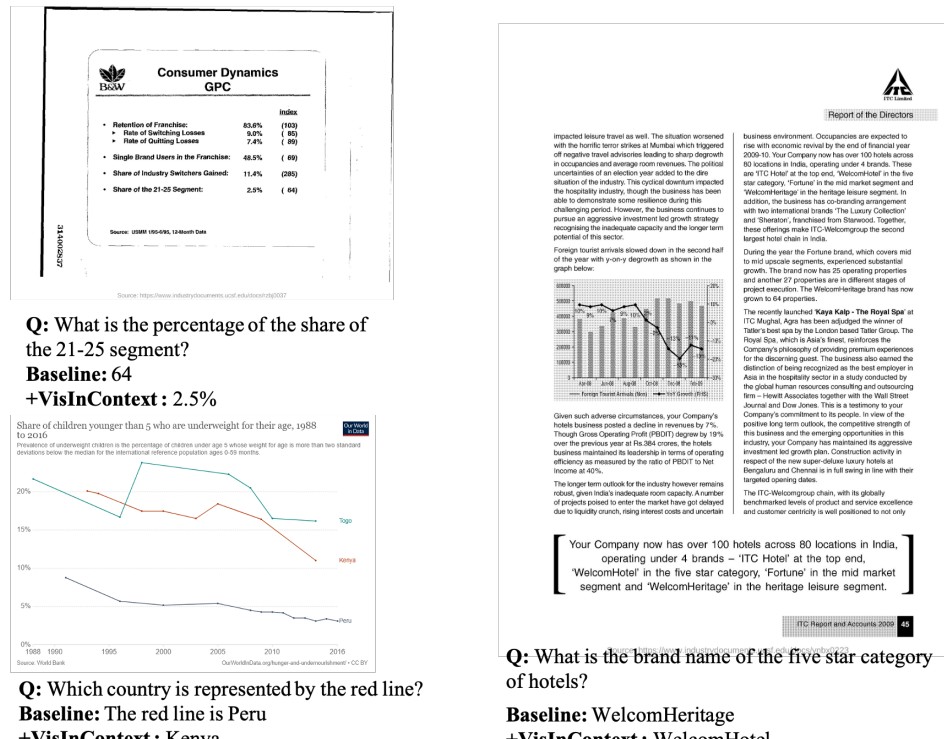

**Q:** What is the percentage of the share of the 21-25 segment?
**Baseline:** 64
**+VisInContext :** 2.5%

**Q:** Which country is represented by the red line?
**Baseline:** The red line is Peru
**+VisInContext :** Kenya

**Q:** What is the brand name of the five star category of hotels?
**Baseline:** WelcomHeritage
**+VisInContext :** WelcomHotel

Figure 7: The document understanding example of our method.

# E    In-Context Few-Shot Example

In this work, we primarily follow the methods from the Flamingo series [1, 14, 9], as these provide comprehensive support for in-context pretraining.

For zero-shot evaluation, the input sequence is formatted as follows:

```
<Visual><Question><Answer>
```

For few-shot evaluation, such as a two-shot evaluation, the input includes two in-context examples. The sequence then becomes:

```
<Visual1><Question1><Answer1><Visual2><Question2><Answer2>
<Visual><Question><Answer>
```

For the text-only input sequence in a text-to-image few-shot setting, the format is:

```
<Question1><Answer1><Question2><Answer2>
<Visual><Question><Answer>
```

Note that we remove all visual tokens to create a longer input sequence, which can then be rendered into a text image for **text-to-image few-shot evaluation**.

# F    Activating the Visual Encoder

One approach involves activating the visual encoder during pretraining, allowing the model to independently learn visual OCR information within the vision encoder.

As shown in Table 13, this method significantly enhances performance in document understanding tasks. However, it also introduces considerable instability during pretraining and requires extended

| Method | DocVQA | | OCR VQA | Classification |
|---|---|---|---|---|
| | val | test | | Hatefulmems |
| **Frozen** | 48.5 | 52.2 | 58.4 | 61.3 |
| **Learnable** | **50.3**(1.9↑) | **54.0**(1.8↑) | **59.0**(0.6↑) | 59.4(1.9↓) |

Table 13: The impact of opening visual encoder during pre-training.

iterations (from 200k to 500k) for convergence. Additionally, it decreases performance in classification tasks. Therefore, we use the frozen visual encoder by default, as the token resampler alone suffices to develop document understanding capabilities.

## G Boarder Impact

This work introduces VisInContext, a method to enhance token efficiency in multi-modality large language models (MLLMs) by using visual tokens to process extended textual contexts.

**Positive Impacts**: VisInContext can democratize access to advanced NLP technologies by reducing computational resources required for long text sequences. This improvement promotes sustainable AI practices by lowering energy consumption and allows researchers with limited resources to utilize powerful MLLMs for applications in education, healthcare, and content generation.

**Negative Impacts**: Potential negative impacts include the misuse of efficient text processing for spreading misinformation or creating deepfake content. Additionally, reliance on visual tokens may introduce biases if training data is not diverse.

Mitigation Strategies: To mitigate these risks, we recommend implementing content moderation, developing ethical AI usage guidelines, and ensuring diverse and balanced training datasets. Continuous monitoring and auditing of AI systems using VisInContext can also help address unintended consequences.

In summary, VisInContext offers significant advancements in token efficiency and computational sustainability, but it is essential to consider and address its broader societal impacts responsibly.

