# OpenReview forum: "Leveraging Visual Tokens for Extended Text Contexts in Multi-Modal Learning"
_NeurIPS.cc/2024/Conference — NeurIPS 2024 poster_

### Official Review · Reviewer_QgLn · 2024-07-01

**Soundness:** 3
**Presentation:** 4
**Contribution:** 4
**Rating:** 6
**Confidence:** 4

**Summary:**

Extending context length is a fundamental challenge for LLMs. Unlike previous approaches that focus on efficiently handling text tokens, this paper introduces a novel method: encoding lengthy text information into image renderings. These image tokens effectively increase the context length and enhance the performance of LLMs across various downstream tasks.

**Strengths:**

The idea is simple yet surprisingly effective. Previous works like PIXEL and CLIPPO suggested treating text as images to eliminate the need for tokenizers and unify various languages into a single image format. In contrast, this paper uses image rendering to encode lengthy texts, employing a PIXEL-like method to enhance long-context understanding.

**Weaknesses:**

1. Information loss from image rendering

The current approach has two potential sources of information loss: 1) rendering long texts into an image, and 2) encoding an image into MLLM embeddings. This loss is not thoroughly investigated in the paper. For instance, Table 2 shows that using the original lengthy 426 tokens outperforms the proposed rendered image, albeit at higher computational costs. While it is acceptable to trade some performance for efficiency, the trade-off should be clearly demonstrated in the paper.

---
2. Optimal compression rate

Due to the potential information loss, rendering very long texts into a single high-resolution image might not be optimal. For a context of, say, 2048 tokens, what is the best approach: a single image with 2048 words, or 32 images with 64 words each? The current ablation study only examines the rendering aspects like font size and font interval threshold. However, the trade-off between the number of images and the number of words per image is a crucial study that should be included in the paper.

---
3. Comparison with other long context methods

A significant weakness of the paper is the lack of comparison with other long context methods. Specifically, the paper must compare the proposed compression-into-image approach with compression-into-text approaches, such as [1-3]. Indeed, text compression loses the original semantics, while image rendering retains all previous words. Therefore, combining both approaches—compressing the semantics first and then rendering them into an image—could potentially offer the best of both worlds.

In addition to the text compression approach, it would be beneficial to discuss the pros and cons of this work in comparison with other long context methods. The current paper only discusses classic efficient self-attention models like Longformer. However, there are more recent and diverse approaches, such as handling long sequences by dividing them into multiple chunks [4-5] or using interpolation positional encodings [6-7], among others.

[1] Mu et al. Learning to Compress Prompts with Gist Tokens. NeurIPS 2023.\
[2] Chevalier et al. Adapting Language Models to Compress Contexts. EMNLP 2023.\
[3] Ge et al. In-context Autoencoder for Context Compression in a Large Language Model. ICLR 2024.\
[4] Bertsch et al. Unlimiformer: Long-Range Transformers with Unlimited Length Input. NeurIPS 2023.\
[5] Song et al. Hierarchical Context Merging: Better Long Context Understanding for Pre-trained LLMs. ICLR 2024.\
[6] Chen et al. Extending Context Window of Large Language Models via Positional Interpolation. arXiv 2023.\
[7] Li et al. Functional Interpolation for Relative Positions improves Long Context Transformers. ICLR 2024.

**Questions:**

This paper claims that long texts can be converted into images. If that's the case, do we need text tokens at all? Can we replace all the text with images, instead of just some prefixes as is currently done?

**Limitations:**

Discussed, but preliminary. The paper only addresses a minor technical limitation regarding static vs. dynamic tokenization of images. However, there are many more potential limitations to consider. For example, I wonder if the current approach would also be effective for larger models, such as Llama-3, which has a longer context length of 8192.

---

> ### Author Rebuttal · Authors · 2024-08-07
>
> **Q1. Information loss from image rendering and trade-off between performance and computation cost:**
>
> **1.Information Preservation in Rendering:**
> The rendering process preserves all the words in the text image, ensuring that no textual information is lost during this step.
>
> **2.Information Encoding:**
> Evaluating potential information loss during the embedding step is challenging as it operates at the feature level.
>
> **3. Performance vs. Computational Cost Trade-off:**
> We acknowledge the trade-off between performance and computational efficiency.  Using the original lengthy 426 tokens can outperform the rendered image for inference, but at a higher computational cost. To further investigate this trade-off, we conducted additional pre-training experiments using the smaller Mistral-7B model. We show the result below and find that with the same in-context text length and batch size, our method shows a slight decrease in performance (e.g., from 25.4 to 24.6 on average). However, our approach allows for `longer in-context lengths and larger batch sizes with the same computational resources`. When utilizing these advantages, the mean accuracy of our method clearly outperforms the baseline.
>
> | **Method**       | **ICL Length** | **BSZ** | **(okvqa)** | **(textvqa)** | **(vizwiz)** | **(vqav2)** | **Caption (coco)** | **Caption (flickr)** | **Mean** |
> |------------------|----------------|---------|------------------|-------------------|------------------|-----------------|--------------------|----------------------|----------|
> | **9B Baseline**  | 512            | 16      | 17.1             | 14.8              | 21.5             | 26.5            | 40.1               | 32.1                 | 25.4     |
> | **+VisInContext**  | 512            | 16      | 16.3             | 15.1              | 20.3             | 25.4            | 39.1               | 31.5                 | 24.6     |
> | **+VisInContext**  | 512            | **64**      | **18.5**             | 17.4              | 22.3             | 27.0            | 41.2               | 31.8                 | 26.4     |
> | **+VisInContext**  | **4096**           | 16      | 18.3             | **19.3**              | **22.5**             | **28.4**            | **42.3**               | **34.8**                 | **27.6**     |
>
> > Caption: We compare the zero-shot evaluation of the OpenFlamingo-9B baseline model with our VisInContext. In this context, BSZ stands for batch size, and the prefix text token length is 128. With the same GPU memory, VisInContext supports a larger batch size or a longer in-context learning (ICL) length.
>
> Due to time limit of rebuttal phase, we only conducted one analysis and will include more detailed discussion on this trade-off in the revised version of our paper.
>
> **Q2. Optimal compression rate:**
> Our current implementation is already multiple text image inputs for longer texts by splitting the text into several images. This allows us to process texts longer than what a single image can accommodate. In Figure 2, we depicted only one image for simplicity and clarity.
>
> The changes in `font size and font interval directly correspond to the number of text images` required. For example, using a smaller font size results in fewer images needed to render the same amount of text. This relationship indicates that adjusting font size and interval settings effectively controls the number of images used.
>
> **Q3. Comparison with other long context methods:**
>
> Thank you for insight comment.
>
> [1]: This method involves an additional fine-tuning stage and modifies the cross-attention mask of the transformer.
> [2]: This method reuses the language model multiple times to obtain a summary vector for each segment and the training pipeline is very different.
> [3]: This approach requires an additional instruction fine-tuning stage on specific instruction data to produce the desired output.
>
> These methods focus on re-use the pre-trained model by fine-tune on specific learning objective[1,2] / instruction data[3]. While these methods are not directly compatible with our pre-train (also increase ICL in this stage) and few-shot evaluation pipeline, these works have inspired us significantly.  We are particularly interested in adopting the strategy from [2] during the inference stage for further increase the ICL. However, the code is hard to modify under our codebase and we are still working on it.
>
> Regarding the pros and cons of other long context methods, we will expand our discussion in the revised manuscript to include:
> [4] and [5]: We will discuss these techniques in lines 287-289. For [6] and [7], we will cover these approaches in lines 290-292, which use positional interpolation to extend the context window.
>
> **Q4: Do we need text tokens?**
> At the current stage, our method still relies on partial text input to perform `next-word prediction and uses autoregressive language modeling loss to optimize the model`. In this way, we can test downstream tasks directly by predicting next word. Completely replacing text tokens with images would require developing a unified learning target beyond the current contrastive loss. This is a complex challenge and an area of ongoing research.
>
> **Q5: If current method suit for larger models such as Llama-3:**
>
> The current approach is indeed suitable for larger models like Llama-3 405B, which has an in-context length of 8192 during pre-training (enabled by techniques like pipeline parallelism on 16 H100 GPUs and low bit quantization). `These techniques are applicable to both vision and language models`, allowing vision tokens to scale similarly.
>
> Furthermore, the parameters of the vision model are only a fraction of the overall model parameters, making it feasible to include more visual tokens without significant overhead. For instance, the Llama-3 405B vision-language model incorporates a ViT-H vision encoder with 630M parameters, demonstrating the compatibility of our method.

---

> > ### Comment · Reviewer_QgLn · 2024-08-10
> > **Response to the Rebuttal**
> >
> > Thank you for the rebuttal. As other reviewers noted, this paper challenges the common belief that "image space typically contains more redundancy than semantic space" and argues that images can serve as better tokenizers than texts. Given the ambitious nature of this claim, it may be more challenging to convince people.
> >
> > However, the proposed concept of rendering images as text opens up several intriguing research directions. For example: 1) What is the optimal way to tokenize the combination of images and text? 2) How can we effectively train large models with these tokens, considering the challenges in next-token prediction?
> >
> > For these reasons, I believe this paper could make a valuable contribution to the conference, sparking new ideas among attendees, and I am inclined to maintain my original rating of acceptance.

---

> > > ### Author Response · Authors · 2024-08-11
> > >
> > > Thanks for your timely and positive comment!
> > >
> > > We want to emphasize that the idea of **challenging the common belief about image redundancy** is `not our primary claim` but rather an interpretation by the other reviewer.  Our discussion is focused solely on rendered text images, which consist of white backgrounds and dark text, rather than real-world images.
> > >
> > > Also, we `do not claim to replace text tokenizers entirely`. Instead, our work explores how rendered text images can be utilized within the scope of multimodal large language models (MLLMs) to increase in-context text length in novel ways.

---

### Official Review · Reviewer_iV3t · 2024-07-12

**Soundness:** 3
**Presentation:** 2
**Contribution:** 3
**Rating:** 4
**Confidence:** 4

**Summary:**

The paper proposes a method to increase the context size of multi-modal large language models. The goal is to increase the context with minimal GPU memory for both training and inference as well as floating pointing operations. Finally, the authors show that the method obtains good results on in-context learning.

**Strengths:**

The paper tackles an important problem and can be of interest to the research community. It seems to achieve promising results and the idea seems interesting, but there are parts that are unclear or need better studying.

**Weaknesses:**

I think that the paper is hard to follow and some parts hard to understand. Please see below for some questions

My other concern is related to the experimental part, that I find a bit weak. The comparison is made against only two existing models and I don't clearly understand the numbers. Firstly, one concern is that there seems to be a degradation in performance with 0 shots in some cases.

Also, probably I am missing something, but let's look at Tab. 1. Wouldn't an increase context size (ICL 256 vs 2048) would allow for more shots to be used? I don't understand what is different in terms of the theoretical context between 0 shot Open-Flamingo and VisInContext. Isn't the same information fed to the model? If yes, then I don't see how the experiment measures the impact of the context size. Or is it just way to feed more information from a particular document as opposed to sub-sampling it? In the later case, I would consider this to be a slightly unfair comparison and I don't really understand what are the benefits of the model if in the zero-shot setup where you have access to a larger portion from the document (as opposed to subsampling) the performance drops quite a lot in some cases (okvqa, vizwiz). This shows that the way that the context is processed is not ideal.

**Questions:**

Questions related to lines 75-76.
1. Both images and text are concatenated to result in a sequence of 256 tokens?
2. Is m chosen so that the concatenation is 256 tokens?
3. What does the corresponding text represent? Based on Fig. 2, `I_i` seems to be an actual image and not an image representing the text, so I am confused. Or the bear image has nothing to do with `I_i`? If it doesn't where do the image tokens `I_i` are used in Fig 2?

Questions lines 81-82.
4. Is this the same m as above? If yes, can you elaborate what the m represents? I assume no, since it's `M` vs `m`, but I think other letters can be used to make this more clear

**Limitations:**

The limitations are briefly discussed

---

> ### Author Rebuttal · Authors · 2024-08-07
>
> **Q1. Questions related to line 75-76.**
>
> _i. Clarification on Token Concatenation:_
> Notice that <visualx> is used as `a placeholder to indicate the position of an image` within the token sequence. It has a `token length of 1`. For example, in a sequence of 256 tokens, the structure could be <visual1><text1><visual2><text2><visual3><text3>, where the visual placeholders are interleaved with text tokens. In this example, 253 tokens out of the 256 are text tokens. If the text exceeds the length limit, it will be truncated during preprocessing.
>
> _ii. Pre-defined Hyperparameter $m$:_
> The hyperparameter $m$ is predefined and set to a specific value, which is 3 in this work. It indicates the number of images included in a sequence. If there are not enough images to meet this value, zero tensors are used as padding to ensure the sequence length remains consistent at 3.
>
> _iii. Explanation of Corresponding Text and Image Tokens:_
> The interleaved dataset, such as MMC4, includes multiple images and their corresponding texts. The "corresponding text" refers to the text paired with a specific image. In our notation, $I_i$ represents the $i$-th raw image. In Figure 2, visual information from both raw images and rendered text images is integrated into the LLM using a `cross-attention mechanism`. In this mechanism, `the text tokens act as queries, while the keys and values are the visual tokens derived from the images, after applying token masking`. Notice that $I_i$ only does self-attention with $T_i$ according placeholder <Visual i>.
>
>
>
> **Q2. Questions about Line 81-82:**
>
> The $M$ in these lines represents the number of rendered text images used.
> It is different from $m$, which is used to denote the number of raw images.
> We will use a different letter in the revised version.
>
>
> **Q3. Do not understand Table 1:**
>
> _1. The details of ICL:_ The evaluation pipeline for MLLMs typically consists of a pre-training stage followed by a fine-tuning or zero-shot evaluation stage on downstream tasks. The "In-context Text Length" mentioned in Table 1 refers to the `pre-training stage (Lines 160-162)`. For downstream tasks, we ensure a fair comparison by maintaining the same number of shots and input settings across all models. This means that the settings for `both the baseline model and our method are identical during downstream evaluation`, allowing us to accurately assess the benefits of longer context pre-training. Our method `naturally supports longer shots during inference (Figure 4)`; however, comparing results between significantly different shot numbers, such as 128 shots versus fewer shots, would be unfair as more shots leads to better result in general. Therefore, we do not include these comparisons in Table 1.
>
> _2. Seems to be a degradation in performance with 0 shots in some cases, okvqa and vizwiz drops a lot :_ Each dataset has its own biases and evaluates different capabilities of the models. It is common for results to be slightly below the baseline in some datasets, but the overall mean performance is significantly better than the baseline. Please refer to Q4 for additional information on RVzob as a supplementary resource regarding the instability associated with increased shots.
>
> **Q4. Comparison against only two models:**
>
> Flamingo and Fuyu are two strong and representative methods: Flamingo represents models with visual encoders, while Fuyu represents models with only linear embeddings. All MLLMs fall into one of these two categories.

---

> > ### Comment · Reviewer_iV3t · 2024-08-10
> > **Rebuttal answer**
> >
> > Thank you for the rebuttal! I confirm I have read the rebuttal and I currently don't have other questions. The rebuttal brings more clarity and answers my questions, but I still think there might be a problem of clarity. Hence, I slightly raise my score.

---

> ### Author Response · Authors · 2024-08-11
>
> Thank you for your prompt feedback and for raising the score! We are glad that our rebuttal brings more clarity and answers all your questions.  We would love to resolve any remaining clarity problems, if you could elaborate more about “there might be a problem of clarity”.

---

> > ### Comment · Reviewer_iV3t · 2024-08-12
> >
> > Hey! Sorry for not being more clear. I am referring to the original questions where I think for most of them, parts of the response from the rebuttal need to be included in the revised paper for added clarity. For example the explanation around Table 1, parameter m and the other parts.

---

### Official Review · Reviewer_zUCF · 2024-07-13

**Soundness:** 3
**Presentation:** 3
**Contribution:** 3
**Rating:** 6
**Confidence:** 5

**Summary:**

This paper introduces a method called Visualized In-Context Text Processing (VisInContext) to address the challenge of processing long in-context texts in multimodal learning, which arises due to the high GPU memory and computational costs associated with lengthy textual content. VisInContext converts long textual content into images and uses a visual encoder to extract textual representations, thereby increasing the in-context text length that can be processed effectively. The method is based on a dual-stream encoder model that employs Token Masking and Text-Centric Contrastive Learning (TCCL) to improve the model's ability to learn from the rendered text images, and the paper demonstrates the effectiveness of VisInContext through experiments on various tasks, showing that it outperforms the baseline in terms of performance and inference cost, while also improving the optical character recognition (OCR) ability of the model.

**Strengths:**

1. The paper is in well-written, which makes it easy to understand.

2. VisInContext can reduce GPU memory usage and FLOPs for both training and inference, allowing the model to handle much longer text contexts with lower computational cost.

3. The model trained with VisInContext delivers better performance on common downstream benchmarks for in-context few-shot evaluation and document understanding.

4. The method shows potential in document understanding tasks, as evidenced by the improvements on DocVQA and OCR VQA datasets, and the enhanced next-word prediction accuracy of the LLM on the Rendered Text dataset.

**Weaknesses:**

1. The proposed method involves multiple steps, including text rendering, token masking, and contrastive learning, which might add complexity to the implementation. This could be a barrier to adoption for some practitioners.

2. From my perspective, the pipeline of "text -> text-rendering -> LLM" seems somewhat derivative from the initial purpose of LLMs (Large Language Models). That is, one may question whether we truly need such a complicated paradigm. If the answer is yes, then this approach appears more akin to an engineering exercise. Further, the performance may be affected by the selected font type, font size, and even the rendered canvas.

3. The process of rendering text into images and then processing these images through a vision encoder may introduce new bottlenecks, particularly as the text length increases. The paper could explore these potential limitations or trade-offs in greater detail.

**Questions:**

refer to weaknesses

**Limitations:**

My primary concern is the necessity of the "text -> text-rendering" process. Perhaps it would be beneficial to explore a new perspective that focuses on aligning visual and textual information more effectively. However, as we know, image space typically contains much more redundancy than semantic space. For the current version, I am not fully convinced that projecting text into the image space is the optimal approach. Furthermore, rendering text may introduce several additional complications that warrant careful consideration.

---

> ### Author Rebuttal · Authors · 2024-08-07
>
> **Q1. Complexity of Implementation and Barrier to Adoption:**
>
> It is not true, and we would like to emphasize that our implementation is not only simple but also easy to adopt for future works.
>
> **i. Text Rendering:** This step is performed during the preprocessing phase on the CPU and can be implemented in OpenCV with just a few lines of code. Specifically, our implementation requires only 5 lines of code.
>
> **ii. Token Masking:** This is a straightforward selection process that can be implemented in a single line of code.
>
> **iii. Contrastive Loss:** The computation of the contrastive loss is performed on the average token representations, which is a standard and easily implemented technique.
>
> Overall, our method is designed to be simple and straightforward, ensuring it is accessible to practitioners without adding undue complexity.
>
>
> **Q2. Derivation from the initial purpose of LLMs; Need for such a complex paradigm; Engineering exercise:**
>
> We disagree with your assessment. Here are the key points:
>
> **1.Clarification on Purpose:** It is not clear what you mean by the "initial purpose" of LLMs. Our focus is on `Multi-modality Large Language Models (MLLMs)`, not solely LLMs. Our VisInContext method significantly increases the in-context text length for MLLMs during both the pre-training and inference stages.
>
> **2. Clarification on why we need such complex paradigm:**  Increasing the in-context text length is a crucial challenge, particularly during pre-training, while most works on LLMs require fine-tuning or post-tuning to support longer contexts. There is no prior work addressing this topic in MLLMs. It is widely recognized that vision encoders are significantly smaller than LLMs. Therefore, processing longer but not highly related text with a visual encoder at a very modest computational cost is an efficient approach. Actually, compared to rendered text images, `text tokenizers in LLM require numerous human-defined preprocessing steps` such as lowercasing, punctuation removal, stop words removal, tokenization, and more—generally involving almost ten steps. In this context, `rendered text images offer a much simpler paradigm` for processing text.
>
> **3. Clarification on engineering exercise:** We use a fixed font and a simple white background. While font size does affect performance, we have already discussed this in our experiments. Our method significantly increases the in-context text length for MLLMs, representing a substantial improvement. This is not merely an engineering exercise but a `strategic enhancement to the model's ability to process long text`. Additionally, it is unclear what you mean by "engineering exercise".
>
> In conclusion, our method addresses a critical need in MLLMs and provides tangible benefits, demonstrating its value beyond a simple engineering exercise.  For MOE based MLLM, we increase the in-context text length from 256 to 2048 during pre-training stage.
>
> **Q3. Potential Bottlenecks of Using a Vision Encoder and Discussion of Limitations:**
>
> We argue that both the LLM and vision encoder have limitations in terms of processing long text sequence, especially for the purpose of extended context for multimodal understanding. Below we summarize the pros and cons of using vision encoder to process long text sequence.
>
> **1. Size Comparison:** The vision encoder in an MLLM is typically much smaller compared to the LLM itself.
> For example, the vision encoder is 340M but the LLM is 56B for MOE-based MLLM.
> This size difference suggests that the vision encoder is a more economical way to process text information.
>
> **2. Efficiency:** Rendered text images can encode more text within the same number of tokens (Lines 98-99), making this method efficient for handling longer contexts.
>
> **3. Complementary Technique:** Our method is complementary to existing techniques for increasing in-context text length in LLMs. It provides an additional means to enhance model performance without replacing current methods.
>
> **4. Limitations and Trade-offs:** We acknowledge potential limitations, such as dynamic token handling, which are discussed in Lines 302-304. However, our experimental results show that the benefits of our approach outweigh this limitation.
>
> **Q4. Concerns about redundancy in image space and the necessity of the "Text -> Text-Rendering" Process:**
>
> We disagree with the notion that projecting text into image space is suboptimal due to redundancy.
>
> **1.Redundancy in Real-World Images vs. Rendered Text Images:** While real-world images often contain redundancy because adjacent patches look similar, include colors, textures,  background noise, which can be redundant and not directly related to the underlying semantic meaning.  But for rendered text image,  since the image is primarily composed of meaningful text with minimal background details, it `closely mirrors the semantic density of the text` itself and significantly `reduces the typical redundancy` found in image space.
>
> In addition, process text with image have following advantages:
>
> **1. Efficiency of Image Tokens:** In our method, the number of image tokens is fewer than the equivalent number of text tokens for the same length of input (Line 98-99). This demonstrates that processing text in image space can be more efficient than using text tokens alone.
>
> **2. Economic Efficiency:** Our approach shows that preprocessing text into image space is more economically friendly compared to relying solely on text models. For example, flops in Figure 1.

---

> > ### Comment · Area_Chair_faAP · 2024-08-13
> >
> > Dear reviewer zUCF,
> >
> > the discussion period draws to a close, could you please check and reply to the authors' response? Please also revise your score accordingly if needed.
> >
> > Sincerely,
> > your AC.

---

> > ### Comment · Reviewer_zUCF · 2024-08-14
> >
> > Thank you for the explanation. I have reviewed the response and comments raised by other reviewers. In my initial comments, the term "derivative" was actually meant to be "deviated". Apologies for the typos.
> >
> > I still have some confusion about the exact helpful clues or information captured by contrastive learning. From the results, I can observe that it truly brings improvement. However, I am wondering what kinds of cases can be improved with and without the TCCL approach. It would be helpful if you could provide some comparison examples, such as cases where TCCL leads to significant improvements versus cases where it does not provide as much benefit. Concrete examples illustrating the strengths and limitations of the TCCL method would give me a clearer understanding.

---

> > > ### Author Response · Authors · 2024-08-14
> > >
> > > Contrastive learning in our method is designed to make the vision encoder and resampler work together as a `"visual text tokenizer."` This means it encourages the embeddings of rendered text images to align with those of regular text tokens, allowing them to capture similar overall meanings. With contrastive learning, we observe:
> > >
> > > **Improved Next Word Prediction Accuracy:** We observed that models trained with contrastive learning `perform better in predicting the next word on the Rendered Text Image dataset`. This improvement indicates that the model has a better understanding of text and stronger OCR (Optical Character Recognition) capabilities. For example, in Figure 3 the validation OCR accuracy (val_ocr_average_acc1) drops significantly from 85.25 to 74.67 when contrastive learning is removed.
> > >
> > > **Enhanced Performance on TextVQA:** According to Table 6, the VisInContext model with contrastive learning shows a significant improvement on the TextVQA dataset, with accuracy increasing from 18.3 to 21.8. This dataset requires the model to `read and understand text within images to answer questions`, highlighting the model's enhanced ability in text-based reasoning when contrastive learning is applied.
> > >
> > > One potential limitation of the model with contrastive loss is its tendency to struggle in scenarios `where the image contains a significant amount of irrelevant or misleading text`, which can lead to incorrect interpretations. To illustrate this, we conducted a simple experiment using images with false text from the Typographic Attack dataset, as discussed in Section 7.2 of [1]. In this experiment, we posed a basic QA task asking, 'What is in this image?' The results showed that **the model with contrastive loss exhibited a higher language model (LM) loss**. For example, when presented with an image of a cup labeled with the misleading text 'iPad,' the model ` incorrectly responded with 'A blue iPad.'`
> > >
> > > Considering we are unable to provide figures at this stage to aid understanding, we believe these examples effectively demonstrate that contrastive learning is particularly beneficial in cases where understanding text within images is essential.
> > >
> > > [1]. Joanna et al, Disentangling visual and written concepts in CLIP,  CVPR'22

---

> > > > ### Comment · Reviewer_zUCF · 2024-08-14
> > > >
> > > > Thanks for the prompt response.  I suppose authors made a good rebuttal and all my concerns have been solved properly. I thus raise my score to WEAK ACCEPT.

---

> > > > > ### Author Response · Authors · 2024-08-14
> > > > >
> > > > > Thank you very much for your thoughtful review and for taking the time to reassess our paper during the rebuttal!

---

### Official Review · Reviewer_Vzob · 2024-07-13

**Soundness:** 2
**Presentation:** 3
**Contribution:** 3
**Rating:** 5
**Confidence:** 4

**Summary:**

The paper introduces Visualized In-Context Text Processing (VisInContext), a novel technique designed to enhance multi-modal learning models by efficiently expanding their in-context text length. This method transforms extensive text into visual tokens, substantially lowering GPU memory consumption and computational requirements during training and inference. Models utilizing VisInContext demonstrate superior performance on standard downstream benchmarks for in-context few-shot evaluation compared to conventional approaches. They also show improved document comprehension, particularly in document QA and sequential document retrieval tasks. An additional advantage of VisInContext is its compatibility with current context-extending techniques, allowing for potential combined applications.

**Strengths:**

- The paper is well-written and easy to follow.
- The suggested method presents an innovative and intriguing concept: converting text into visual representations to decrease computational expenses.

**Weaknesses:**

- The motivation of “text-only in-context few-shots experiment” is not clear. These experiments appear tailored specifically to validate the proposed method rather than addressing practical applications. In particular, the use of text-only versions of visual question answering (VQA) or image captioning tasks for in-context learning seems questionable. The relevance and applicability of such text-only adaptations of inherently visual tasks in this context require further justification.
- There are some unconvincing parts about token masking. In Line 87~88, the paper says the masking ratio of raw image tokens is 1.0. Then the model does not observe the raw image at all. Or, is the model initialized with OpenFlamingo Model, especially for cross-attention layer and resampler? If so, how does the token masking probability affect the model’s ability to learn text semantics from visual inputs? For my intuition, it seems there could be some trade-off of partly observing (masking) raw-image and learning text-image semantics from rendered text images at the same time.
- Minor :
    - Reference, Figures needs compile check. There are some errors. (Line 159, 521, etc. )
    - The formats of Table 1, 2 need to be more polished

**Questions:**

- Table 1 exhibits an interesting trend where the baseline model occasionally shows improved performance with an increased number of shots in certain scenarios. This pattern might be attributed to specific characteristics of the datasets or peculiarities of the classification task at hand. What explanation do the authors propose for this counterintuitive observation?

**Limitations:**

The paper addresses thier limitation.

---

> ### Author Rebuttal · Authors · 2024-08-06
>
> **Q1. The motivation of “text-only in-context few-shots experiment” is not clear:**
>
> The motivation behind the “text-only in-context few-shots experiment” aligns with the common practices in mainstream few-shot learning architectures like Flamingo, IDEFICS, and EMU2[1]. These models often include `text-only prompts for zero-shot evaluation`.
>
> For example, in a typical zero-shot evaluation setting, the input to the model is structured as __<text 1><text 2><text 0><image 0>__, where "0-shot" actually involves using 2-shot text-only prompts (Lines 170-173). This design helps bootstrap the output format, enabling the model to `produce answers that follow the style of the prompts`. Follow this line, we test the longer prompt beyond 2-shot text data.
>
> So the primary reasons for using text-only prompts in these experiments are:
> 1. **Testing Prompt Understanding**: This setup tests the ability of model to clearly understand the prompt and follow instructions.
> 2. **Practicality**: It provides a practical advantage when prompt images are difficult to obtain, ensuring the model can still perform well in the absence of visual input.
>
> Motivated by these, we propose to evaluate text-only in-context few-shot and demonstrate that text rendered as images can be comparably effective as raw text.
>
> **Q2. The details about token masking in Lines 87-88:**
>
> There seems to be a misunderstanding. The vision feature input to the cross-attention model is a sum of raw image tokens and rendered text image tokens. As depicted in Figure 2.
>
> In our implementation, the `raw image tokens are masked with a pre-defined probability`, which is 50% in our case (details are provided in Lines 107-110). This means the model only observes the raw image tokens half of the time. The reference to a 1.0 masking ratio in lines 87-88 indicates that we mask all the raw image tokens when the model is not supposed to see them at all.
> However, the rendered text image tokens remain intact and still observed by the model.
>
> The rationale behind this approach is that we experimentally found that models _tend to overly rely on raw pixel images, often ignoring the information from rendered text images_. By masking the raw image tokens, we encourage the model to pay attention to the rendered text images, thereby learning text-image semantics more effectively. For inference, we sum raw image token and rendered text image token (Line111-112).
>
> **Q3. Some other minor typos:**
>
> Thank you for pointing out these minor issues. We will fix the reference and figure compilation errors (Lines 159, 521) and polish the formats of Tables 1 and 2 to ensure they meet the required standards.
>
> **Q4. Explanation for Baseline Model Occasionally Performing Better:**
>
> Attention to Visual Tokens: For classification tasks, the model might prioritize distinguishing visual tokens over textual information. This could lead to better performance in scenarios where visual cues are more prominent.
>
> ` More shots do not always result in better performance.` As seen in related works like Flamingo (Table 1 in this work), it's not uncommon for more shots to occasionally result in worse performance. This variability can be due to the _random selection of example shots from the support sets_. If the selected examples are quite different from the query image-text pair, the model's performance might drop. Therefore, it is normal that the baseline outperforms our work occasionally and we `mainly focus on mean accuracy over all datasets.`
>
> Some works, like RICES [2], analyze this phenomenon and focus on prompt ensembling by retrieving similar samples from the query set to highly improve multi-shot performance, which is not our focus.
>
> [1]. Sun Q, Cui Y, Zhang X, et al. Generative multimodal models are in-context learners[C]//Proceedings of the IEEE/CVF Conference on Computer Vision and Pattern Recognition. 2024: 14398-14409.
>
> [2]. Yang Z, Gan Z, Wang J, et al. An empirical study of GPT-3 for few-shot knowledge-based VQA. Proceedings of the AAAI Conference on Artificial Intelligence, 2022, 36(3): 3081-3089.

---

> > ### Comment · Reviewer_Vzob · 2024-08-11
> > **Response to the rebuttal**
> >
> > Thanks for the response. I carefully read through the authors' responses and the discussions among the other reviewers. While most of my initial concerns have been addressed, I still have an unclear part, which is related to the token masking. In lines 87-88, the paper states that "which ensures that the model won’t simply be ignoring the text images during training." However, according to the rebuttal, "a 1.0 masking ratio" is needed for the situation when the model is not supposed to see the raw image tokens, such as text-only in-context few-shot, or zero-shot evaluation. The word "training" in line 88 confuses me. Could you elaborate on this more, please?

---

> > > ### Author Response · Authors · 2024-08-11
> > >
> > > Thanks for your timely response! The term "Training" in lines 87-88 refers to the **application of token masking exclusively during the pre-training stage**.
> > > During the pre-training stage, token masking is applied to ensure the LLM sees only the raw image 50% of the time. For the remaining 50%, the LLM processes both the rendered text image and the raw image together.
> > > This approach _prevents the model from adopting a trivial solution that completely disregards the rendered text image_.
> > >
> > > During `downstream evaluation tasks, no tokens are masked`. For instance, in the in-context text-only few-shot setting in Table 2, where rendered text images are available, the raw image token remains unmasked.
> > > The vision feature input to the cross-attention model in this scenario is the sum of the raw image token and the rendered text image token.
> > >
> > > For zero-shot evaluation in Table 1, where no rendered text image is present, only the grey-shaded components in Fig. 2 are retained.
> > >
> > > I hope this clears up any confusion. Please let me know if further clarification is needed.

---

> > > > ### Comment · Reviewer_Vzob · 2024-08-13
> > > >
> > > > I still don't get it. The masking ratio is set to 0.5 for pretraining, and no tokens are masked during downstream evaluation. Then, why do we need lines 87~88, where the masking ratio is set to 1.0, if we don't face the situation where "the model is not supposed to see them at all"?

---

> ### Author Response · Authors · 2024-08-13
>
> The masking ratio of 1.0 mentioned in Lines 87-88 `differs from` the predefined probability of 50% referenced in Lines 107-110.
>
> The predefined probability of 50% in Lines 107-110 indicates that, during pre-training, there is a **50% chance that the raw image tokens will be masked**.
>
> The masking ratio of 1.0 in Lines 87-88 refers to the `specific implementation of masking`, where all raw image tokens are masked.
>
> The purpose of Lines 87-88 is to address the issue where combining tokens from raw images and text images directly caused the network to overlook the text-image input (as mentioned in Lines 105-107).

---

> > ### Comment · Reviewer_Vzob · 2024-08-14
> > **Final rating**
> >
> > Ok, I got the context. Nevertheless, I think the manuscript should be clearer. Overall, I still think the idea of using rendered text is interesting and underexplored, despite the debate on the necessity or complexity of using rendered texts. Therefore, I will keep my initial rating.

---

### Decision · Program_Chairs · 2024-09-25

**Decision:**

Accept (poster)

**Comment:**

The paper originally received mixed reviews, with concerns including:

1. clarity of presentation
2. motivation and applicability in practice
3. potential limitations and trade-offs due to occluding parts of the image
4. method complexity
5. comparison to additional baselines

Following rebuttal, observing the author-reviewer discussions, the AC got the impression most concerns were addressed, with only some concerns of clarity of presentation remaining for reviewer iV3t who also suggested a constructive solution to resolve the clarity issue in the revised manuscript.

In light of the above, the AC recommends accepting the paper and urges the authors to incorporate all the discussions into the revised manuscript. Especially the constructive suggestion by iV3t.